# Self-Assembly Motifs of Water in Crystals of Palladium *β*-Amino Acid Complexes Influenced by Methyl Substitution on the Amino Acid Backbone

**David B. Hobart** †, **Vraj G. Patel, Heather Pendergrass, Jacqueline Florio and Joseph S. Merola** *

Department of Chemistry, Virginia Tech, Blacksburg, VA 24061, USA; dhobart@vt.edu (D.B.H.); vrajgp94@vt.edu (V.G.P.); heathp2@vt.edu (H.P.); jacquelineflorio@gmail.com (J.F.)

* Correspondence: jmerola@vt.edu; Tel.: +1-(540)-231-4510
† Current address: Air Liquide Advanced Materials, Branchburg, NJ 08876, USA.

**Abstract:** Amino acid complexes of transition metals show interesting hydrogen-bonding motifs. In this paper, the syntheses and structures of three *β*-amino acid complexes of palladium that differ only by the substitution on the *β*-carbon will be discussed. With only hydrogen on the *β*-carbon, no additional water is incorporated into the crystal lattice and hydrogen-bonding is all complex-to-complex. With the addition of one and two methyl groups on the amino acid *β*-carbon, water is incorporated into the crystal lattice giving intricate water networks held together by complex-to-water and water-to-water hydrogen-bonding networks.

**Keywords:** palladium; amino acid; metal complex; hydrogen bonding; crystal lattice

## 1. Introduction

Many research teams [1–3] including ours [4–7] have been studying the metal complexes of amino acids, to examine both their catalytic activity as well as their biological activity. Perhaps one of the most influential investigators in this area is Wolfgang Beck whose series "Metal Complexes of Biologically Important Ligands" contains over 175 publications in a research career spanning over 50 years [1]. A great many of those publications deal with alpha-amino acid complexes of a wide variety of metals. Quite appropriately for this paper, one of Beck's more recent publications concerns Pd and Pt complexes of long chain amino acids [8].

Much less studied, however, are the *β*-amino acid complexes of metals. This is somewhat surprising since a beta-amino acid should form a stable 6-membered ring chelate with metals, coordinating through the carboxylate O atom and the amino N atom. There are a number of *β*-amino acid complexes of first-row transition metals such as Co and Cu [9–16] but many fewer examples of heavier metals such as Pd and Pt [17–19]. It is not surprising that the hydrogen-bonding (H-bonding) capabilities of amino acids play a large role in the solid-state structures of the their metal complexes. In an iridium valine complex on which we reported, H-bonding sets up a unique helical arrangement of molecules that extends along a $4_3$ screw axis of the space group P4$_3$, with L-Valine and D-Valine generating crystals with opposite helical turns in P4$_1$ [4,5].

Relevant to the issue of H-bonding between molecules is the bonding of water contained in crystalline lattices. Waters of hydration are a common occurrence in purely inorganic salts [20] but the exact nature of water molecules contained in organic or organometallic crystals is often complex [21,22]. Infantes et al. [23]

have mined the examples of water in crystal lattices and have developed a nomenclature for describing these water motifs. It is now possible to search for water motifs in crystals using a feature of the CCDC program, Mercury [24]. Fully understanding all of the factors in directing a particular hydrogen-bonding motif is still an important ongoing topic of study [25–33]. Hydrogen-bonding as a tool in crystal engineering is also an increasingly important area [26,34].

Recently, we have been investigating the structures and catalytic properties of amino acid complexes of palladium. We have published a paper on glycine and substituted glycines [35] as well as a paper on palladium complexes of proline and proline homologs [36]. A common and consistent feature of all complexes is their propensity for hydrogen bonding—with the carbonyl oxygen acting as H-bond acceptor and the NH$_2$ hydrogens acting as H-bond donors. In many cases, crystallization readily occurs only with water molecules in the lattice creating some unusual H-bonding networks.

In this paper, we discuss the structures of three Pd *β*-amino acid compounds and how 0, 1 and 2 methyl groups on the amino acid backbone greatly influence the H-bonding, the inclusion of water and the overall crystal lattice.

## 2. Results and Discussion

### 2.1. Complex Syntheses

Three different, readily available *β*-amino acids were examined: 3-aminopropanoic acid, (S)-3-aminobutanoic acid and 3-amino-3-methylbutanoic acid. Syntheses of the palladium compounds were straightforward and the procedures followed those we have reported on previously [35]. In terms of isomer formation, 3-aminopropanoic acid gave exclusively the trans isomer, trans-bis(3-aminopropanoate) palladium, compound **1**. The thermal ellipsoid plot for **1** is shown in Figure 1. The $^1$H and $^{13}$C nuclear magnetic resonance (NMR) spectra also show signals indicative of only one isomer in solution. This complex was the subject of a previous paper by Krylova et al. [19] and a room temperature crystal structure of trans-bis(3-aminopropanoato)palladium was reported. Krylova and co-workers also reported that the cis-isomer could be synthesized by heating the trans isomer to 80° followed by cooling to 0° but no X-ray crystal structure was obtained and we could not reproduce that finding.

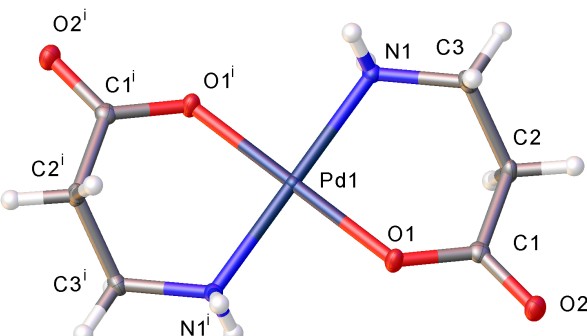

**Figure 1.** Thermal ellipsoid plot of trans-bis-(3-aminopropanoato)palladium, complex **1**. Atoms marked with superscript i are generated by the inversion operation. Ellipsoids are shown at 50% probability.

(S)-3-aminobutanoic acid gave predominantly the trans isomer of bis-[(S)-3-aminobutanoato] palladium, compound **2** but, in solution, approximately 25% of the cis isomer was observed by NMR spectroscopy. However, despite the 75/25 solution ratio of trans to cis observed in solution, all crystal growing attempts yielded solely the cis isomer. The thermal ellipsoid plot of the cis-isomer of compound **2** is shown in Figure 2.

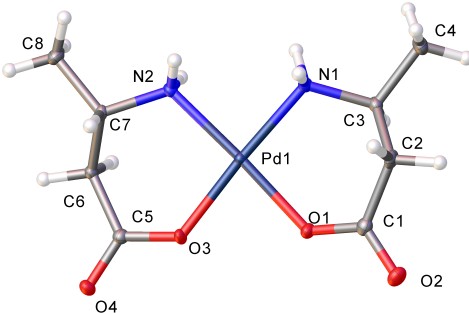

**Figure 2.** Thermal ellipsoid plot of cis-bis-((S)-3-aminobutanoato)palladium, complex **2**. Lattice waters are not displayed in this figure. Ellipsoids are shown at 50% probability.

The reaction between 3-amino-3-methylbutanoic acid and palladium(II) acetate yielded exclusively the trans isomer of [bis-[3-amino-3-methylbutanoato]palladium, compound **3**, as evidenced by NMR spectroscopy. The single crystal X-ray structure also shows only the trans isomer, Figure 3.

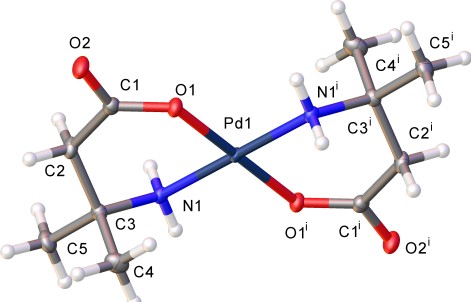

**Figure 3.** Thermal ellipsoid plot of trans-bis-(3-amino-3-methylbutanoato)palladium, complex **3**. Lattice waters are not displayed in this figure. Ellipsoids are shown at 50% probability.

Gaussian 09 DFT calculations support that the trans isomers are more stable than the cis (see Supplementary Materials) but the results do not explain why some cis isomer was observed in solution for the (S)-3-aminobutanoic acid product and why only the cis product was found in the solid state. Of course, ΔG calculations provide guidance only if the system is at equilibrium. Even if the system were at equilibrium, a lower solubility for the cis isomer of the (S)-3-aminobutanoic acid product could explain why crystallization favors that product. It is also a possibility that the hydrogen-bonding network established in the lattice for the cis (S)-3-aminobutanoic acid product favors crystallization of that isomer even if solubility were not an issue.

Aside from the different isomer possibilities, the inclusion of waters of hydration and the hydrogen-bonding motifs those waters form are very interesting. First, trans-bis-(3-aminopropanoato) palladium crystallizes with no waters of hydration even though the solution from which it was derived was 50% water by volume. The intermolecular interactions in this crystal lattice are derived solely from complex-to-complex hydrogen bonding interactions of the form N−H···O where both hydrogen atoms on the amino nitrogen are involved in donating to oxygens in two different adjacent complexes. Each hydrogen atom is involved in two different forms of H-bonding. The first N-H donates to a carbonyl oxygen of the carboxylate (or, to phrase it differently, to the carboxylate oxygen that is NOT bonded to the palladium) with an H···O distance of 2.09 Å. The second N-H bond appears to be partly H-bonded to both carboxylate oxygen atoms in an adjacent complex with H···O bond distances of 2.37 Å each. We presume

that this complex with the simplest β-amino acid has no steric issues with bringing the complex molecules close together enough to satisfy N–H⋯O bonding without inclusion of waters of hydration.

Addition of a methyl group to the β-carbon atom of the amino acid (S)-3-aminobutanoic acid yields two major differences when the amino acid is used to form a palladium(II) complex. First, the cis, rather than the trans complex is formed. This is somewhat surprising in that the addition of a methyl group would simplistically be expected to favor the trans complex since the methyl group would provide greater steric repulsion in the cis compound. At the moment, the only speculation we would make is that the cis compound is less soluble and thus crystallization favors formation of the cis isomer. Of course, this requires that the cis and trans be in equilibrium in solution and that does not appear to be the case, so solution dynamics and crystallization dynamics need to be studied further.

The second major difference is that the complex molecules are not able to approach each other closely enough to form N–H⋯O bonds between complex molecules. Instead, several waters of hydration are included in the lattice and the N–H⋯O bonding is exclusively complex N-H to water oxygen binding as well as water O-H to non-bonded carboxylate oxygen of the complex (as well as to oxygen in other water molecules in the lattice.) Overall, the lattice picture is best described as alternating channels of complex and of water.

The amino acid with 2 methyl groups attached to the β-carbon, 3-amino-3-methylbutanoic acid, now yields solely (by NMR spectroscopy) the trans structure, a result that makes sense based on the increased steric repulsion that placing the 4 methyl groups on the same side of the palladium would involve.

Interestingly, now that the $NH_2$ groups are on opposite sides of the molecule, there is a trajectory along which the N-H hydrogen atom can approach closely enough to hydrogen-bond to the non-bonded carboxylate oxygen atom of an adjacent complex molecule. However, only one of the N-H hydrogen atoms can satisfy its donating ability in this manner. The other donates to waters of hydration in the crystal lattice. The combined features of the line of approach of the complex molecules as well as H-bonding to waters of hydration yields a complex H-bonding motif and an overall picture that, simplistically, can be described as layers of complex interspersed with layers of water.

*2.2. Hydrogen Bonding Motifs*

Compound **1** does not contain any lattice waters as mentioned above. But the hydrogen-bonding network is interesting nonetheless. Figure 4 shows a portion of the packing diagram for compound **1** indicating that there are four channels of one orientation of the molecule surrounding one channel of molecules at approximately right angles to the four with N–H⋯O bonding holding the complexes together.

Once a methyl group replaces a hydrogen on the β-carbon, the crystal lattice now includes water. Figure 5 shows a thermal ellipsoid plot viewed down the a axis of the crystal lattice highlighting the hydrogen bonding. From this view, it would appear that the crystal lattice may best be described as two-dimensional channels of complex interspersed with two-dimensional channels of water. Using the CCDC program Mercury and its CSD -Materials tools [24] the water space calculation shows that the crystal, by volume, is 20 % water. Also using Mercury, Figure 6 was generated and even more clearly illustrates the channels of water in the crystal lattice.

When *both* hydrogen atoms on the β-carbon of the parent β-alanine are replaced by methyl groups, even more water is incorporated into the crystal lattice. Figure 7 shows a portion of the packing diagram for compound **3** where it can be seen that layers of complex alternate with layers of water. Mercury analysis of water space more clearly illustrate the alternating layers of water and complex molecules (Figure 8). That same analysis shows that the crystal lattice for compound **3** is 38% water by volume. Lourdes et al. have mined the CCDC database for different motifs of water hydrogen-bonding in crystal structures and have developed different designators for water motifs [23]. For complex **3**, the motif would be described

as T4(2)6(2). T4(2)6(2) indicates an infinite tape of alternating 4,6-membered rings of oxygen atoms sharing one edge. The "(2)" designator tells that the point of connection between the rings is two oxygen atoms. Figure 9 shows just the water in the crystal water layer with the 4,6-membered ring tape highlighted. The figure also shows that where two of the infinite tapes abut, 5-membered oxygen rings form.

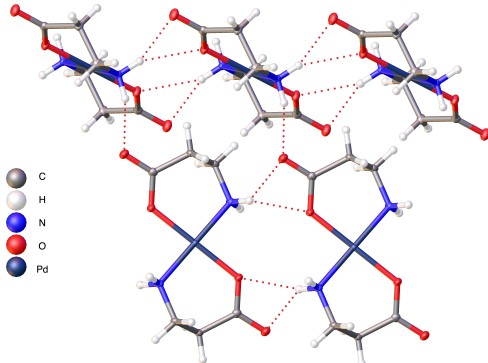

**Figure 4.** Thermal ellipsoid plot of complex **1** showing relative orientations of hydrogen bonding in the lattice (red dotted lines). Ellipsoids are shown at 50% probability.

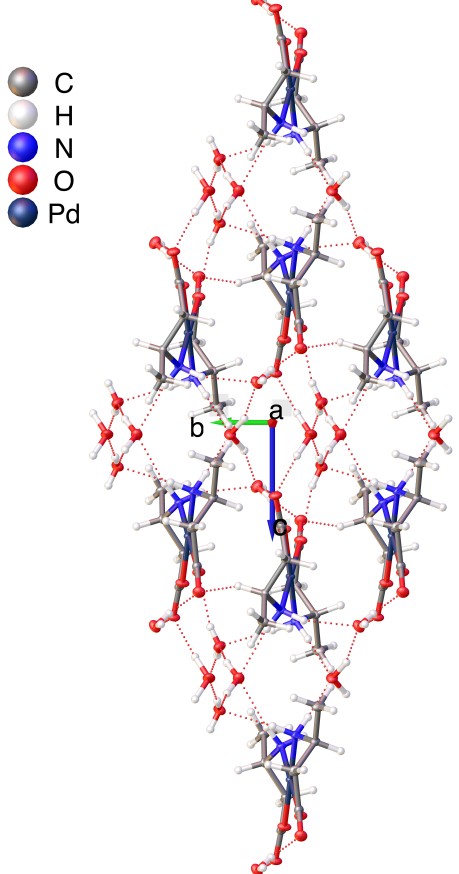

**Figure 5.** Thermal ellipsoid plot of complex **2** viewed down the a axis showing relative orientations of hydrogen bonding in the lattice. Ellipsoids are shown at 50% probability.

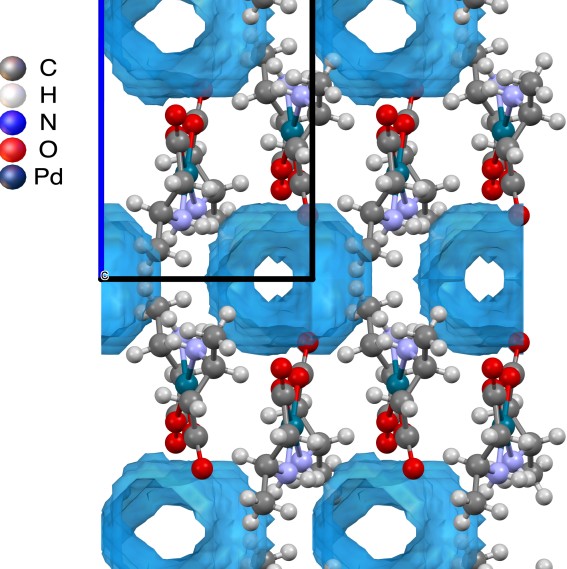

**Figure 6.** Water space analysis of complex **2** crystal lattice viewed down the a axis showing the water channels in the crystal.

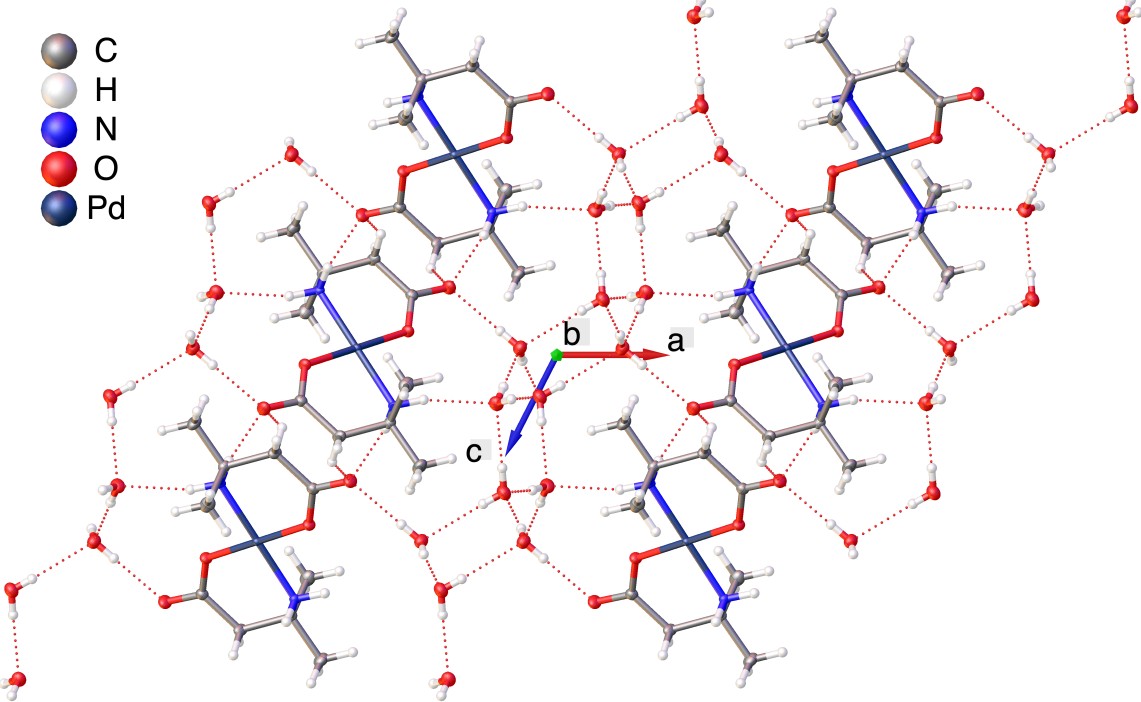

**Figure 7.** Thermal ellipsoid plot of a portion of the packing diagram of complex **3** viewed down the b axis showing alternating layers of water and complex molecules.

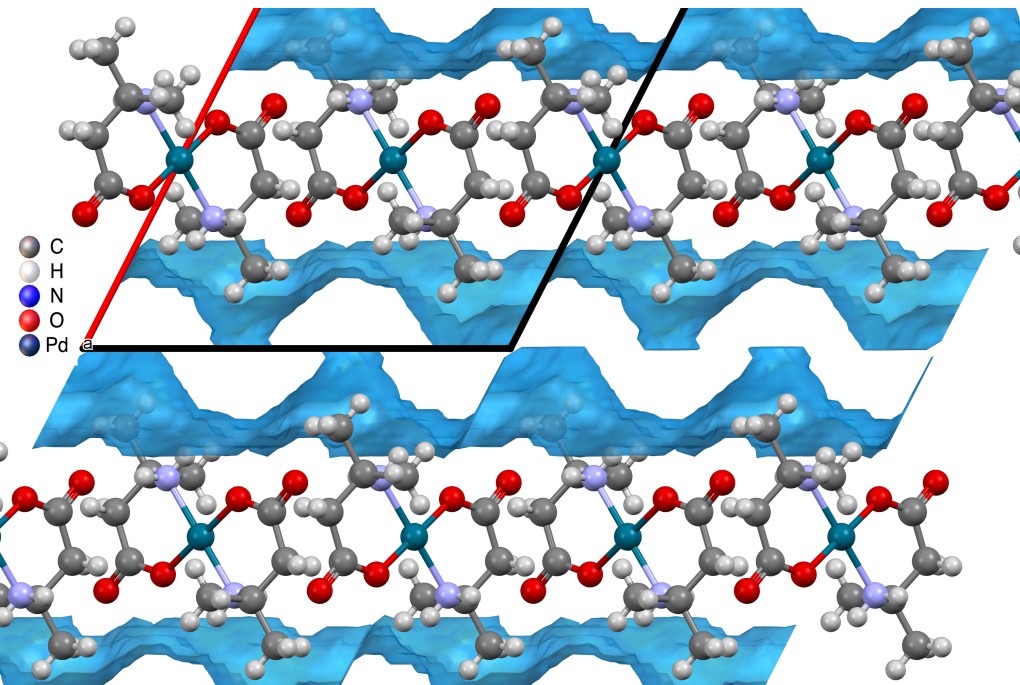

**Figure 8.** Water space calculation plot of complex **3** viewed down the b axis showing alternating layers of water and complex molecules.

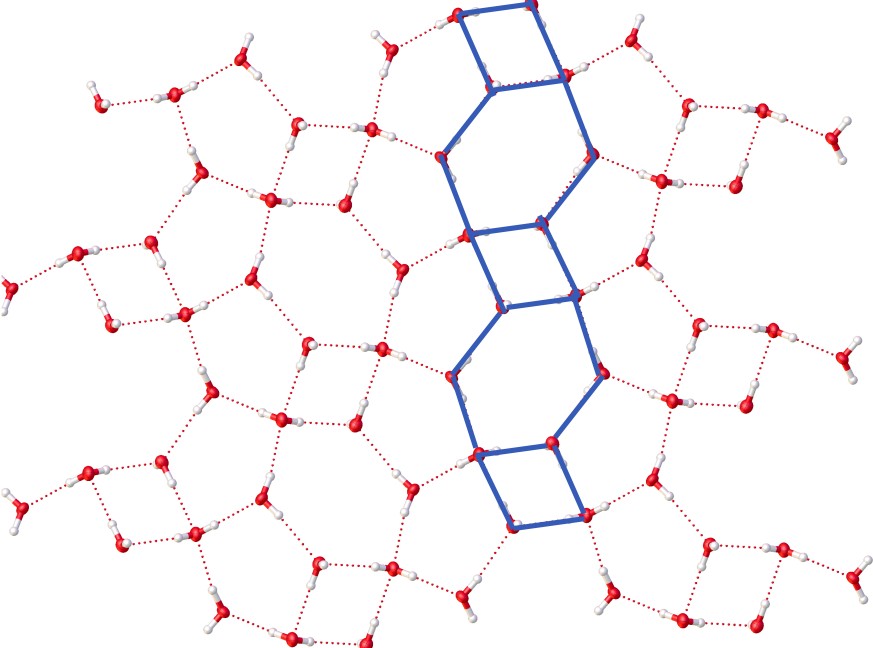

**Figure 9.** Water motif of complex **3** in the crystal lattice water layer. Red spheres are oxygens, white spheres are hydrogens and dotted red lines show hydrogen bonding. The repeating T4(2)6(2) motif is highlighted in blue.

## 3. Conclusions

In this paper, we have described the synthesis of 3 different palladium complexes of readily available *β*-amino acids and have shown that the presence of 0, 1 or 2 methyl groups on the *β* carbon atom greatly affects the crystal lattice. No methyl groups (3-aminopropanoic acid) forms a palladium complex with a trans arrangement of the amino acid ligands with no inclusion of water in the crystal lattice. One methyl group (S-3-aminobutanoic acid) forms complexes with palladium with evidence for both cis and trans isomers. However, only the cis isomer formed suitable crystals for X-ray diffraction and water was included in the crystal lattice. In this case, the water is present in the lattice in one dimensional channels. 3-amino-3-methylbutanoic acid (2 methyl groups on the *β*-carbon atom) appears to form only the trans isomer and water is included in the crystalline lattice in the form of an infinite layer between layers of complex.

Future work will examine further modifications that can be made to *β*-amino acids [37–39] and the effect that those modifications will have on crystalline lattice formation. For example, N-substitution [18] will not only affect the steric demands of the amino acid ligands but also change the number of hydrogen bond donors. Changing the type of substitution on the *β*-carbon (phenyl, longer chain alkyl) should also affect how the complexes pack in the lattice and how water can pack as well. This palladium-*β*-amino acid system affords many possibilities for engineering the crystal lattice and, hopefully, developing consistent principles for hydrogen-bonding motifs.

## 4. Experimental

### 4.1. General Synthetic Techniques

All compounds are air-stable and water soluble, so no special handling techniques are needed. The *β*-amino acids were purchased from commercial sources and used as received. Palladium(II) acetate was purchased from Pressure Chemical (Pittsburgh, PA, USA).

### 4.2. Synthesis of Complexes

#### 4.2.1. Synthesis of trans-bis-(3-aminopropionato)palladium(II)

A four dram vial was fitted with a magnetic stir bar and charged with palladium(II) acetate (51.1 mg, 0.2276 mmol) and 2.5 mL of 50/50 (*v*/*v*) acetone/water. The mixture was stirred until all solids had dissolved. To this was added *β*-alanine (44.6 mg, 0.5008 mmol) and stirred overnight. The reaction solution turned from a clear red-orange to a clear pale-yellow supernatant with a pale-yellow precipitate. The supernatant was transferred via pipette to a clean vial and allowed to evaporate to give clear yellow rectangular prisms. The pale-yellow precipitate was washed with cold water and dried under vacuum. The combined yield of single crystals and precipitate was 59.6 mg of product (0.2109 mmol, 93% yield). Trans-Pd($C_3H_6NO_2$)$_2$ was identified on the basis of the following data: $^1$H NMR (400 MHz, $D_2O$) *δ* 2.54–2.44 (m, 2H), 2.42–2.32 (m, 2H). HRMS/ESI+ (*m/z*): [M + H]$^+$ calcd for Pd($C_3H_6NO_2$)$_2$, 282.9910; found, 282.9904.

#### 4.2.2. Synthesis of cis-bis-((S)-3-aminobutanoato)palladium(II)

A four dram vial was fitted with a magnetic stir bar and charged with palladium(II) acetate (53.2 mg, 0.2370 mmol) and 2.5 mL of 50/50 (*v*/*v*) acetone/water. The mixture was stirred until all solids had dissolved. To this was added (S)-3-aminobutanoic acid (53.8 mg, 0.5213 mmol) and stirred overnight. The reaction solution turned from a clear red-orange to a clear pale-yellow supernatant with a pale-yellow precipitate. The supernatant was transferred via pipette to a clean vial and allowed to evaporate to give

clear yellow rectangular prisms. The pale-yellow precipitate was washed with cold water and dried under vacuum. The combined yield of single crystals and precipitate was 68.6 mg of product (0.2208 mmol, 93% yield). Cis-Pd($C_4H_8NO_2$)$_2$ was identified on the basis of the following data: $^1$H NMR (400 MHz, D$_2$O) $\delta$ 3.62 (h, *J* = 6.7 Hz, 1H), 2.53–2.48 (m, 1H), 2.43–2.36 (m, 1H), 1.32–1.19 (m, 3H). HRMS/ESI+ (*m/z*): [M + H]$^+$ calcd for Pd($C_4H_8NO_2$)$_2$, 311.0223; found, 310.9069.

### 4.2.3. Synthesis of trans-bis-(3-amino-3-methylbutanoato)palladium(II)

A four dram vial was fitted with a magnetic stir bar and charged with palladium(II) acetate (57.5 mg, 0.2561 mmol) and 3.0 mL of 50/50 (*v/v*) acetone/water. The mixture was stirred until all solids had dissolved. To this was added of 3-amino-3-methylbutanoic acid (63.1 mg, 0.5386 mmol) and stirred overnight. The reaction solution turned from a clear red-orange to a clear pale-yellow supernatant with a pale-yellow precipitate. The supernatant was transferred via pipette to a clean vial and allowed to evaporate to give clear yellow rhombohedral prisms. The pale-yellow precipitate was washed with cold water and dried under vacuum. The combined yield of single crystals and precipitate was 78.1 mg of product (0.2306 mmol, 90% yield). Trans-Pd($C_5H_{10}NO_2$)$_2$ was identified on the basis of the following data: $^1$H NMR (400 MHz, Methanol-d4) $\delta$ 2.38 (s, 2H), 1.31 (s, 6H). HRMS/ESI+ (*m/z*): [M + H]$^+$ calcd for Pd($C_5H_{10}NO_2$)$_2$, 339.0536; found, 339.0473.

### *4.3. X-ray Crystallography*

For compounds **1**, **2** and **3**, crystals suitable for X-ray diffraction formed by allowing the slow evaporation of the reaction solution. Data was collected on an Oxford diffraction Gemini_Mo_Eos diffractometer with crystals maintained at 99.9 K during data collection. Structures were solved using SHELXT [40] and refined using SHELX [41] software and figures and other crystallographic output were made using OLEX2 [42]. Analyses of water in the crystal lattices of complexes **2** and **3** were carried out with the program Mercury [24].

1. **Crystal Data for compound 1, $C_6H_{12}N_2O_4Pd$** (M = 446.77 g/mol): monoclinic, space group P2$_1$/n (no. 14), a = 5.6938(3) Å, b = 8.8497(4) Å, c = 9.0421(5) Å, $\beta$ = 104.686(6)°, V = 440.73(4) Å$^3$, Z = 2, T = 97(5) K, $\mu$(MoK$_\alpha$) = 2.090 mm$^{-1}$, Dcalc = 2.129 g/cm$^3$, 4467 reflections measured (7.68° $\leq$ 2$\theta$ $\leq$ 64.158°), 1451 unique (R$_{int}$ = 0.0315, R$_{sigma}$ = 0.0356) which were used in all calculations. The final R$_1$ was 0.0448 (I > 2$\sigma$(I)) and wR$_2$ was 0.0636 (all data).

2. **Crystal Data for compound 2, $C_{16}H_{44}N_4O_{15}Pd_2$** (M = 745.35 g/mol): monoclinic, space group I2 (no. 5), a = 10.7953(2) Å, b = 7.38120(10) Å, c = 17.8370(3) Å, $\beta$ = 97.217(2)°, V = 1405.33(4) Å$^3$, Z = 2, T 100.15(5) K, $\mu$(MoK$_\alpha$) = 1.352 mm$^{-1}$, Dcalc = 1.761 g/cm$^3$, 14648 reflections measured (7.462° $\leq$ 2$\theta$ $\leq$ 64.882°), 4716 unique (R$_{int}$ = 0.0381, R$_{sigma}$ = 0.0421) which were used in all calculations. The final R$_1$ was 0.0292 (I > 2$\sigma$(I)) and wR$_2$ was 0.0556 (all data).

3. **Crystal Data for compound 3, $C_{10}H_{32}N_2O_{10}Pd$** (M = 446.77 g/mol): P2$_1$/c (no. 14), a = 12.4228(5) Å, b = 7.0497(2) Å, c = 12.5509(5) Å, $\beta$ = 117.080(5)°, V = 978.67(7) Å$^3$, Z = 2, T 97(5) K, $\mu$(MoK$_\alpha$) = 0.993 mm$^{-1}$, Dcalc = 1.516 g/cm$^3$, 9102 reflections measured (6.932° $\leq$ 2$\theta$ $\leq$ 64.77°), 3258 unique (R$_{int}$ = 0.0379, R$_{sigma}$ = 0.0474) which were used in all calculations. The final R$_1$ was 0.0448 (I > 2$\sigma$(I)) and wR$_2$ was 0.0636 (all data).

## 5. Associated Content

### *Accession Codes*

CCDC 1916293–1916295 contain the supplementary crystallographic data for this paper. These data can be obtained free of charge via www.ccdc.cam.ac.uk/data_request/cif, or by emailing

data_request@ccdc.cam.ac.uk, or by contacting The Cambridge Crystallographic Data Centre, 12 Union Road, Cambridge CB2 1EZ, UK; fax: +44 1223 336033.

**Supplementary Materials:** The following are available at http://www.mdpi.com/2073-4352/9/11/590/s1, Table of results from DFT calculations, $^1$H NMR spectra for all 3 compounds, high resolution mass spectra for all 3 compounds and the complete experimental information and results on bond distances and angles from the three X-ray structures.

**Author Contributions:** Conceptualization, J.S.M.; Formal analysis, D.B.H. and J.F.; Investigation, D.B.H., H.P. and V.G.P.; Project administration, J.S.M.; Resources, J.S.M.; Supervision, D.B.H. and J.S.M.; Writing—original draft, D.B.H. and J.S.M.

**Funding:** This research received no external funding.

**Acknowledgments:** The authors thank the Hamilton Corporation for a generous donation of syringes and Christine DuChane for aid in obtaining NMR spectra for compound **2**.

**Conflicts of Interest:** The authors declare no conflict of interest.

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
