# Peer review of "Self-Assembly Motifs of Water in Crystals of Palladium β-Amino Acid Complexes Influenced by Methyl Substitution on the Amino Acid Backbone"

_crystals, doi:10.3390/cryst9110590_

Round 1
Reviewer 1 Report
This article presents the synthesis of 3 Palladium complexes with amino acids. Authors clearly indicate the presence of a cis-trans equilibrium as well as the role of a substituent (methyl group) on the presence or absence of structural water in the solid crystal. The findings are original and well supported either by experimental evidences and theoretical calculations. Thus, I consider it suitable for publication after some minor details.
Cis-trans equilibrium is mentioned but not drawn in any scheme, it will be ideal to present it in figure 1 (for example). The first sentence of the article is somehow forced, I rather prefer something like: “Many research, teams including ours, have been studying metal complexes of amino acids” SI It will be ideal if all NMR had the same scale (from -1 to 12 ppm). Then magnification can be introduced as an inset. Background of spectra is non homogeneous (compound 1 and 3 have grid). SI format for x-ray data seem different, borders, position in the page…
Author Response
I have changed the first sentence, hopefully for the better. While I see merit in the suggestion of the NMR spectra having similar backgrounds and scales, there is no real benefit of direct comparison of the NMR spectra, so I will leave the SI as it is. I am not clear about the comment regarding the X-ray data, but I will try to make all of the SI as clear and consistent as possible.
To accommodate both this reviewer and reviewer 4 concerning cis-trans isomerism, the discussion in the paper has been altered to down-play solution issues and to focus on the solid state structures.
Reviewer 2 Report
Hobart et al. report the synhesis and crystal structures of three palladium complexes with differently substituted β-aminocarboxylates. They emphasize the distinct crystal packing in the three structures, which exhibit very different hydrogen bonding patterns. All this is discussed in detail and the observations are rationalized. The discussion is speculative in parts but this is then made also very clear. The writing is eloquent, very clear, has a red line going through and is, hence, rather pleasant to read. The entire work is scientifically sound and perfectly suitable for Crystals.
Only few, and minor issues should be revised prior to publishing:
Line 21: ref 18 reports the exact same complex 1, as does this submission. This could be pointed out more clearly.
In line 30 there is one “in” too many.
In Figure 4 only one type of H-bonding is shown while two types are discussed. Could the Figure be changed to illustrate both?
The sentence starting in line 137 “No methyl group…” and the following ones have a strange wording. Even though putting it this way might be word-economical I find it difficult to digest. An option would be to start with “The ligand with no methyl group…” instead.
Line 183: either delete the “of” or add the amount in front of the “of”.
Author Response
I thank this reviewer for the careful reading and kind words.
I have made all of the changes suggested by this reviewer.
Line 21: ref 18 reports the exact same complex 1, as does this submission. This could be pointed out more clearly.
I have edited this sentence to make it clear that we know that there was a previous room temperature structure.
In line 30 there is one “in” too many.
Deleted extra "in"
In Figure 4 only one type of H-bonding is shown while two types are discussed. Could the Figure be changed to illustrate both?
I have altered figure 4 to better make this point.
The sentence starting in line 137 “No methyl group…” and the following ones have a strange wording. Even though putting it this way might be word-economical I find it difficult to digest. An option would be to start with “The ligand with no methyl group…” instead.
I have corrected this sentence to "
The β-amino acid with no methyl groups on the backbone (3-aminopropanoic acid) forms a palladium complex with a trans arrangement of the ligands with noinclusion of water in the crystal lattice.
Line 183: either delete the “of” or add the amount in front of the “of”.
I have deleted the "of"
Reviewer 3 Report
The manuscript "Self-Assembly Motifs of Water in Crystals of Palladium _-Amino Acid Complexes Influenced by Methyl Substitution on the Amino Acid Backbone." by Ae Kyung Hobart et al. describes characterization and structural determination of three compounds. The topic of the article fits the scope of Crystals and will be interesting for its readership.
But to evaluate the scientific part is very difficult because:
As can be found inline 200: «CCDC 1916293–1916295 contains the supplementary crystallographic data for this paper». Unfortunately there are no records with such numbers in the database (ccdc.cam.ac.uk/data_request/cif) ! There are not any tables with experiment parameters, parameters of crystal structures refinement, interatomic bonds or atoms coordinations in the text of paper. Thus it is not possible to assess the quality of a single crystal experiment. The authors in the article discuss hydrogen bonds but do not give tables of their values! The figures are not clear. There is not any legend in figures captions.I propose to make changes to the text of the article and solve the problem with CCDC numbers. If it will be done, then I can evaluate the scientific part.
Author Response
Reviewer states: "As can be found inline 200: «CCDC 1916293–1916295 contains the supplementary crystallographic data for this paper». Unfortunately there are no records with such numbers in the database (ccdc.cam.ac.uk/data_request/cif) ! ! There are not any tables with experiment parameters, parameters of crystal structures refinement, interatomic bonds or atoms coordinations in the text of paper. Thus it is not possible to assess the quality of a single crystal experiment"
All of the experimental data and X-ray parameters are in the supplementary information so I am sorry that the reviewer didn't see it there. In addition, for a structure deposited pre-publication, one needs to search differently: there is a place to ask for the structure "pre-publication for referee purposes."
Perhaps, in the future, in addition to depositing structural data to CCDC, you could also request .cif and cifcheck files be uploaded for refereeing purposes.
Also, H-bond tables are also in the supplementary for each structure. Personally, I like a lot of the detailed data such as these to be in supplementary where they do not hinder the flow of the narrative, but are nonetheless available. In this era where many journals, such as "Crystals" are online, the ability to access the main body as well as the supplementary is trivially easy.
However, I do note that I did not include a simple table in the main body of the paper as requested in instructions to authors and I have fixed that omission.
This reviewer also asked for legends in the figure captions.
Legends for figures where the atoms are not already labeled are now included
Reviewer 4 Report
This manuscript reported by Joseph S. Merola and coworkers described crystal structures of palladium β-amino acid complexes. It is rather surprising that crystal structures of these simple complexes have not yet been reported to the best of my knowledge. I recommend this manuscript for publication in Crystals.
The main purpose of the present study is to investigate hydrogen-bonding networks in the crystal lattice. The authors have successfully shown that subtle modification of the β-amino acid structure induces the incorporation of water molecules in the crystal lattice. But it is somewhat difficult to grasp the importance of the present achievement for non-expert readers. I suggest the authors to explain the background of crystal engineering in more detail in the introduction part.
My only concern is a cis-trans isomerism in solution. In the case of (S)-3-aminobutanoic acid, the authors observed major and minor 1H NMR signals, which are assigned as trans and cis isomers, respectively. But this assignment is not validated, because the present manuscript lacks solution structural studies such as NOE experiments. Furthermore, only one set of 1H NMR signals for the complexes of 3-aminopropanoic acid and 3-amino-3-methylbutanoic acid does not necessarily mean a single isomer in solution, because a fast exchange between cis and trans isomers could give such a 1H NMR spectra. More studies such as variable-temperature 1H NMR are required to address this issue.
In my opinion, it is better to remove the description regarding the cis-trans isomerism in solution, which is not necessary for the main purpose of the present manuscript. I recommend the authors to just describe the observation on 1H NMR spectra without assigning cis and trans isomers.
Author Response
This reviewer does indeed bring up a valid point concerning cis-trans isomerism. I cannot ignore it completely, but I will change the discussion solely to deal with the isomers in the crystal structures and some discussion of the relative energies from Gaussian calculations.
Round 2
Reviewer 3 Report
Experimental data and details of structure solving look good. Now, I can recommend publish this article.